# Financial health as a measurable social determinant of health

**Emily Brown Weida** [1]*, **Pam Phojanakong**[1], **Falguni Patel**[2], **Mariana Chilton**[1]

**1** Department of Health Management and Policy, Dornsife School of Public Health, Drexel University, Philadelphia, Pennsylvania, United States of America, **2** Children's Hospital of Philadelphia, Philadelphia, Pennsylvania, United States of America

* eab364@drexel.edu

## Abstract

### Objectives

Financial health, understood as one's ability to manage expenses, prepare for and recover from financial shocks, have minimal debt, and ability to build wealth, underlies all facets of daily living such as securing food and paying for housing, yet there is inconsistency in measurement and definition of this critical concept. Most social determinants research and interventions focus on siloed solutions (housing, food, utilities) rather than on a root solution such as financial health. In light of the paucity of public health research on financial health, particularly among low-income populations, this study seeks to: 1) introduce the construct of financial health into the domain of public health as a useful root term that underlies other individual measures of economic hardship and 2) demonstrate through outcomes on financial, physical and mental health among low-income caregivers of young children that the construct of financial health belongs in the canon of social determinants of health.

### Materials and methods

In order to extract features of financial health relevant to overall well-being, principal components analysis were used to assess survey data on banking and personal finances among caregivers of young children who participate in public assistance. Then, a series of logistic regressions were utilized to examine the relationship between components of financial health, depression and self-rated health.

### Results

Components aligned with other measures of financial health in the literature, and there were strong associations between financial health and health outcomes.

### Practice implications

Financial health can be conceived of and measured as a key social determinant of health.

**Funding:** This work received funding from the following sources of financial support: Robert Wood Johnson Foundation, W.K. Kellogg Foundation, PEW Charitable Trusts, Annie E. Casey

Foundation, First Hospital Foundation, Claneil Foundation, Inc. The funders had no role in study design, data collection and analysis, decision to publish, or preparation of the manuscript.

**Competing interests:** The authors have read the journal's policy and have the following competing interest: This study was funded by the Claneil Foundation, Inc. This does not alter our adherence to PLOS ONE policies on sharing data and materials.

## Introduction

Exposure to prolonged economic hardship has detrimental impacts on individual health and well-being [1–8]. Despite the widespread acceptance of economic hardship as a social determinant of health, the way it has been measured is restricted to income in relation to the federal poverty line (FPL) or to issues related to housing insecurity, food insecurity, healthcare trade-offs, and other measures related to tangible basic needs [2, 9–12]. Comprehensive solutions, such as efforts to improve financial capability (known as a combination of self-efficacy, skill, attitude, and knowledge needed to make financial decisions) [13], build assets to protect from financial shocks, and reduce income volatility are missing from public health research [14–17]. As well, interventions tend to be "siloed." That is, they address access to safe and affordable housing, medical care, and food in distinct interventions without addressing financial health, or the longer-term and underlying issues of having enough money, income, or financial capability [3, 18, 19]. Financial health is a comprehensive assessment of finances that includes the ability to support meeting basic needs, which also encompasses opportunities to save and build wealth. Building on research on associations between income and physical health [8, 20, 21], this work demonstrates that financial health is an important, independent social determinant of health that can be defined, measured, and influenced to improve health and wellbeing.

### Financial health in the literature

Although the concept of financial health (or financial well-being) has been discussed for several decades, most literature on the subject lies outside the public health domain, and definitions (if present) vary from source to source. Often, if a definition is provided, such as in gray literature, details on the measurement or operationalization of this concept are scarce, or the measurement is relating to a narrow aspect of a family's financial situation such as income level or employment. Despite this, some recent studies have tied specific measures of financial health with health outcomes [6–8, 19, 21]. For example, several researchers found associations between financial *distress* or *well-being* and perceived health [22–24] and mental health outcomes [25]. Additionally, several studies have examined links between health and financial *literacy* [26], as well as financial *stress* and mental and physical health outcomes. However, the majority of these studies lie outside of public health journals, and definitions or measurements of 'financial health' [27–29], 'financial well-being' [25, 30, 31], 'financial well-ness' [32], or 'financial fitness' [33] are inconsistent across studies. This poses challenges, such that the domains where most of the financial health studies are located (i.e. consumer finance) often do not target interventions for lower-income populations or under/un-banked populations. Moreover, they do not create opportunities for interdisciplinary interventions aimed at improving physical and mental health.

Experiences of economic hardship such as housing and food insecurity are symptoms of a deeper hardship in financial health. Fig 1 depicts how the authors conceptualize how financial health is related to other forms of economic insecurity (housing insecurity, food insecurity, energy insecurity) such that it is a *root* cause of these other insecurities. Efforts to create meaningful opportunities to increase wealth in order to improve food security, housing security, and other economic hardships have been limited by the lack of consistent, validated measures and definitions of individual financial health [31, 34–37].

For instance, Netemeyer et al. [38] identified two dimensions of financial "wellbeing" (characterizing the dynamic relationship between finances and well-being): *behaviors* like managing the day-to-day and planning, and *perceptions* such as "a state of being wherein you have control over day-to-day, month-to-month finances [. . .] have the financial freedom to make the choices that allow one to *enjoy* life." The Center for Financial Services Innovation (CFSI)

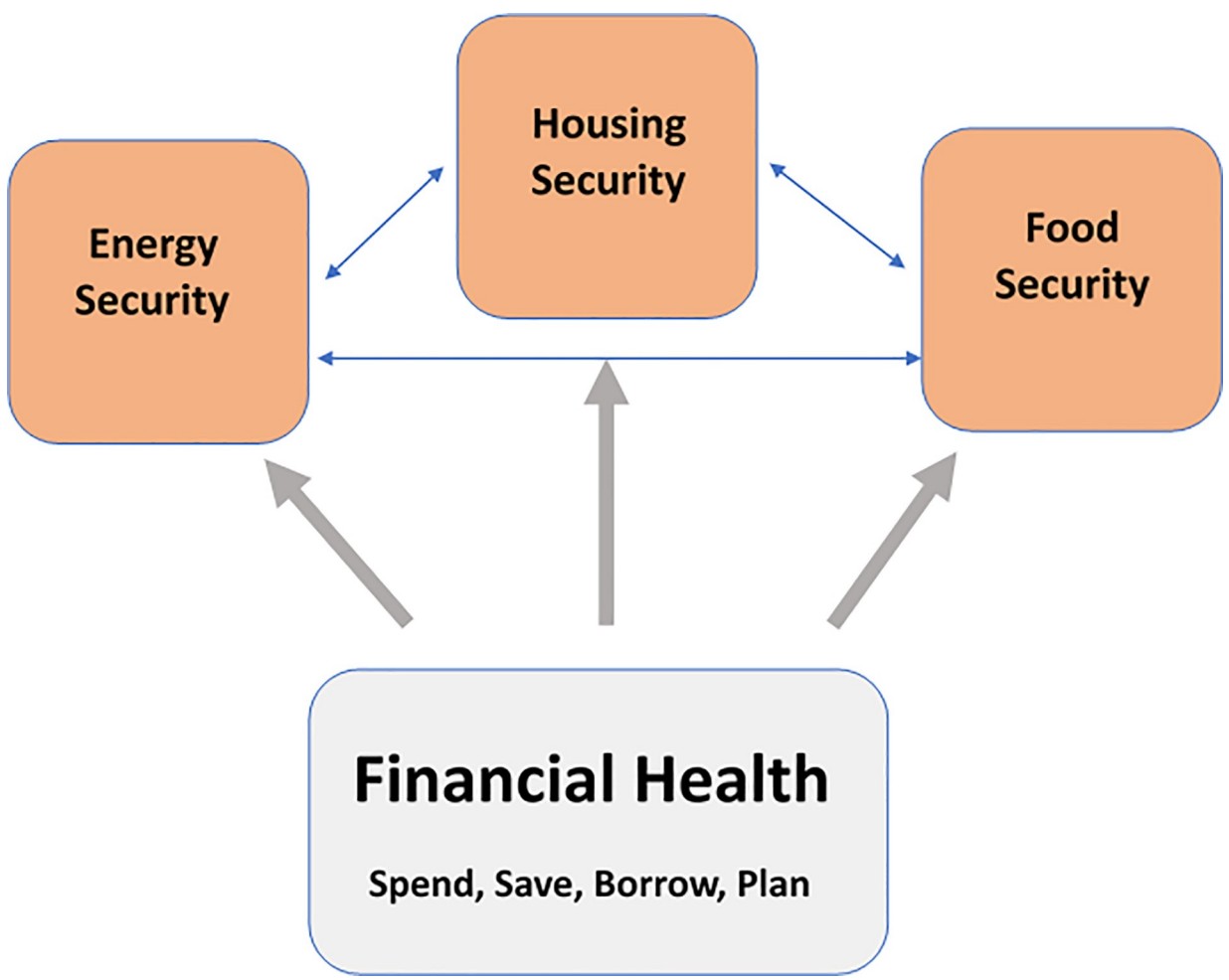

**Fig 1. Financial Health underlies other economic social determinants.**

established a definition and measure of financial health based on a mixed-methods study examining financial habits (cash flow, spending, income volatility, savings, investments, etc.) among moderate-income families [39], stating financial health consists of effective management of day-to-day financial decisions; resilience in facing "ups and downs"; and ability to seize opportunities for financial security and mobility" [40].

In light of the paucity of public health research on financial health, particularly among low-income populations, this study seeks to: 1) introduce a construct of financial health into the domain of public health defined by more than just individual measures of economic hardship, and 2) demonstrate via models on financial, physical and mental health among low-income caregivers of young children that the construct of financial health belongs in the canon of social determinants of health.

## Methods

### Study sample

The Building Wealth and Health Network (The Network) began as a randomized controlled trial in 2014 (N = 103), operating out of Drexel University in Philadelphia, PA. The Network continued as a single arm intervention study (Phase II; N = 373); a trauma-informed peer

support financial empowerment program to address depression and economic hardship through building social capital and financial capability among caregivers of children aged 6 years or younger. All participants in The Network were referred to as "members" to reflect the peer format nature of the program to foster social capital, connection and belonging. Network members attended 16 Financial SELF Empowerment Group sessions on financial topics including building and fixing credit, reducing debt, and encouraging entrepreneurship, while also creating a culture of healing through peer support around issues of safety, emotional management, loss and letting go, and developing a sense future. Members received support in opening a savings account with a partner credit union. Members responded to surveys at baseline, and every three months for a year about economic securities, employment, finances, entrepreneurship, depression, and child health. All members provided written informed consent. Full methods and outcomes [41–44] of the Network can be found elsewhere [45]. The Drexel Institutional Review Board approved this study.

### Economic hardship and financial measures

As mentioned above, the Center for Financial Services Innovation (CFSI) developed a definition and measure for financial health for consumer finance in 2016. The CFSI identifies four components of financial heath that mirror daily financial activities: *spend*, *save*, *borrow* and *plan*. Indicators include paying bills on time and in full (*spend*), having sufficient long-term savings (*save*), having a prime credit score (*borrow*) and planning ahead for expenses (*plan*). CSFI components draw from banking and finance industry standards (e.g.: having a debt-to-income ratio below 36%, or a particular credit score) [40]. Given its comprehensive scope, the parameterization of financial health was guided by the CFSI. Finance-related questions were selected from The Network's survey that matched face validity with the CFSI's categories of *spend*, *save*, *borrow*, and *plan*, as well as their suggested survey questions [40]. This analysis focused on ability to pay bills (*spend*), having checking and savings accounts (*save*), borrowing money (*borrow*), and financial planning behaviors (*plan*) (Table 1). Two spending questions (e.g. "Over the past month, would you say your family's spending on living expenses was less

**Table 1. Mapping of CSFI indicators with the Building Wealth and Health Network's (The Network) survey questions.**

| Domain | CSFI Indicators | The Network Survey Questions |
|---|---|---|
| **Spend** | Spend less than you earn<br>Pay bills on time and in full | - Over the past month, would you say your family's spending on living expenses was less than its total income? (Yes = 0, No = 1)<br>- Over the last 2 months, have you paid a late fee on a loan or bill? (Yes = 0, No = 1)<br>- In the last three months, has the utility company shut off your utilities for not paying bills? (Yes = 0, No = 1)<br>- In the last 3 months were there any days that the home was not heated or cooled because you couldn't pay the bills? (Yes = 0, No = 1) |
| **Save** | Have sufficient living expenses in liquid savings<br>Have sufficient long-term savings or assets | - Do you have a savings account? (Yes = 0, No = 1)<br>- Do you have a checking account? (Yes = 0, No = 1) |
| **Borrow** | Have a sustainable debt load<br>Have a prime credit score | -Within the last three months, I repaid the money I owed on time. (Yes = 0, No = 1)<br>-Do you owe anyone money? (please include banks, friends, family, payday lenders, etc) (Yes = 0, No = 1) |
| **Plan** | Have appropriate insurance<br>Plan ahead for expenses | -Do you currently have at least one financial goal? (Yes = 0, No = 1)<br>-Do you currently have a personal budget, spending plan or financial plan? (Yes = 0, No = 1)<br>-Do you currently have an automatic deposit or electronic transfer set up to put money away for a future use? (Yes = 0, No = 1) |

than its total income?", "Over the last 2 months, have you paid a late fee on a loan or bill?") are from the Financial Behavior, Knowledge and Self-Efficacy Scale [46], which assesses respondents' financial behaviors, knowledge and feelings of efficacy around financial decision-making, while the other two spending questions (e.g. "In the last three months, has the utility company shut off your utilities for not paying bills?" "In the last 3 months were there any days that the home was not heated or cooled because you couldn't pay the bills?") are taken from a validated energy insecurity screener, but used in this analysis as indicators for inability to pay bills [47]. These spending questions were chosen as they aligned with the CFSI's spending indicators (pay bills on time and in full, and spend less than you earn) as well as their suggested survey questions [40]. Questions relating to savings and checking accounts (e.g. "Do you have a savings account" and "Do you have a checking account") were chosen as they aligned with savings [40]. Borrowing questions (e.g. "Do you owe anyone money?" and "In the last three months I repaid the money I owed on time") aligned with CFSI's *borrow* components and come from previous work [48] and Danes [46], respectively. The financial planning questions (e.g. "Do you currently have at least one financial goal?", "Do you currently have a personal budget, spending plan, or financial plan?", "Do you currently have an automatic deposit or electronic transfer set up to put money away for a future use?") are from the Financial Behavior, Knowledge and Self-Efficacy Scale [46]. These questions were chosen as they matched face validity with the CFSI "planning" questions [40]. Affirmative answers scored one point, negative answers scored zero (e.g.: for "Do you have a savings account?," a yes was scored 1, a no was scored 0). Some questions were scored in reverse in order to align behaviors across categories (e.g.: for the *borrow* question, "Within the last 3 months, I paid the money I owed on time" a yes was scored 0, a no was scored 1).

Energy insecurity was assessed by a validated scale [47] asking respondents to answer whether in the last 3 months: the gas/electric company sent a letter threatening to shut off service whether energy service was not delivered, the home was not heated/cooled, and whether the cooking stove was used to heat the home because the family could not pay heating bills. Housing insecurity was assessed with a validated scale [9, 49, 50] that measures access to adequate and stable housing, where housing insecurity is indicated by affirmative response to at least two of the following: overcrowding (more than two people per bedroom) or multiple moves (two or more moves in the previous year). Household food insecurity (HFI) within the last 30 days was assessed using the validated 18-item Household Food Security Survey Module [51].

## Health and depression measures

Self-rated physical health was assessed with a question adapted from The National Health and Nutrition Examination Survey [52], where health is self-rated as "excellent," "good," "fair," or "poor," then categorized into excellent/good and fair/poor. Self-reported depressive symptoms were assessed using the Center for Epidemiological Studies-Depression Revised 10 (CES-D-10) [53], a ten-item screening tool validated to assess risk for clinical depression with good sensitivity, specificity and high internal consistency [53]. The CES-D-10 measures depressive symptoms over the previous week: 0 = rarely or none of the time (<1 day), 1 = some or a little of the time (1–2 days), 2 = occasionally or a moderate amount of the time (3–4 days), and 3 = most or all of the time (5–7 days). The range of the 10-item scale is 0 to 30 and the recommended cutoff score of ≥10 was used to indicate presence of depressive symptoms.

## Statistical analysis

For this analysis, the full Phase II baseline sample (N = 373) consisting of participants recruited between 2015–2017 was used. Members that responded to the finance-related questions

described above, economic hardship (food insecurity, energy insecurity and housing insecurity), as well as metrics on self-rated health and depressive symptoms, were included in the analysis. To aggregate over the range of the original 10 survey variables comprised of economic hardship and financial questions (discussed above, see Table 2), principal components analysis (PCA) was used to reduce the variables into a smaller number of "dimensions" [54]. This approach was utilized because it captures the most variance in the data as examining each variable individually may not be sufficient to differentiate financial health levels. Additionally, simply adding up the responses to each question assumes that all responses should be weighted equally when they may have differential impacts on financial health. To avoid collinearity due to the interrelated nature of energy security, housing security and food security, only questions relating to energy security were used for the PCA. Variables were considered meaningful if the absolute value of factor loadings were greater than 0.35 and components were only considered if their eigenvalues were >1.00 [55].

To test associations between financial health components from the PCA and self-rated health and depression, separate multivariate logistic regressions were run modelling the odds of poor-fair vs. good-excellent self-rated physical health and depression (yes/no) with each of the financial health components, adjusting for marital status, age, race/ethnicity, education, employment and other established and related social determinants of health: food, housing and energy insecurity. Results were considered significant at α = 0.05. Tests were run in SAS 9.4 and R.

## Results

Demographic information is provided in Table 2. Over 90 percent of respondents were Black/African American women and participating in some form of public assistance such as The Temporary Assistance for Needy Families (TANF) program or The Supplemental Nutrition Assistance Program (SNAP). Less than a third of respondents (26.7%) had a savings account at baseline and fewer than half (36.9%) had a checking account. Notably, more than half of the respondents were food insecure and over a third were severely housing insecure.

The four financial health components—"can't pay bills" (*spend*), "having assets" (*save*), "owing money" (*borrow*), and "planning" (*plan*), identified from The Network survey using the CSFI framework are presented in Table 3. Their associated eigenvalues are 1.28 (*spend*), 1.97 (*save*), 1.36 (*borrow*), 1.25 (*plan*), accounting for 29%, 20%, 19%, and 18% of variance in the original data, respectively, or 86% of the variance in the original data combined. Individual survey items and their loading factor values are listed in Table 3. The higher the factor loading of a given financial item, the greater the contribution of that item to the component. The *spend* component was characterized by experiencing a utility shutoff (factor loading = 0.651) and having no heat in the home (factor loading = 0.598). The *save* component was characterized by having a checking account (factor loading = 0.411), savings account (factor loading = 0.424), and an automatic deposit set up (factor loading = 0.378). The *borrow* component was characterized by having paid a late fee on a bill (factor loading = 0.569) and owing money (factor loading = 0.568). The *plan* component contained at least one financial goal (factor loading = 0.667) and a personal budget (factor loading = 0.551). No survey items overlapped between components (Table 3).

Table 4 shows results from the logistic regression models examining the association between having significant depressive symptoms with each of the aforementioned financial health components, as well as the association between fair/poor physical health and each of the financial health components. All displayed odds ratios are adjusted for demographic variables and food, housing and energy insecurity. A higher score in owing money (*borrow)* was

Table 2. Baseline characteristics of Building Wealth and Health Network members, the Network Phase II, 2015–2017 (N = 373).

| Variable | N (%) |
|---|---|
| **Demographics** | |
| Caregiver age (Mean, SD) | 28.0 (11) |
| No. of children in household (Mean, SD) | 2.08 (1.28) |
| **Gender** | |
| Male | 19 (5.1) |
| Female | 354 (94.9) |
| **Race/Ethnicity** | |
| Black | 341 (91.4) |
| White | 9 (2.4) |
| Hispanic | 13 (3.5) |
| Other | 10 (2.7) |
| **Partner in home** | 79 (21.3) |
| **Sexual Orientation** | |
| Heterosexual | 318 (85.2) |
| Gay or lesbian | 13 (3.5) |
| Bisexual | 28 (7.6) |
| **Education** | |
| Some high school | 94 (25.3) |
| High school or GED | 172 (46.1) |
| Some college | 106 (28.3) |
| **Recruitment Site** | |
| County Assistance Office | 258 |
| Community | 115 |
| **Financial Measures** | |
| **Checking Account** | 138 (36.9) |
| **Savings Account** | 100 (26.7) |
| **Currently Employed** | 65 (17.3) |
| **Receive TANF** | 267 (77.0) |
| **Receive SNAP** | 358 (96.8) |
| **Receive WIC** | 203 (54.9) |
| **Household food security** | |
| High | 112 (30.2) |
| Marginal | 63 (16.9) |
| Low | 92 (24.8) |
| Very low | 105 (28.3) |
| **Housing Insecurity** | |
| Secure | 143 (38.4) |
| Moderate | 101 (27.1) |
| Severe | 129 (34.7) |
| **Energy Insecurity** | |
| Secure | 240 (64.5) |
| Moderate | 42 (11.3) |
| Severe | 91 (24.4) |
| **Health Measures** | |
| **Self-rated health** | |
| Excellent, very good, good | 223 (60.1) |
| Fair or poor | 148 (39.9) |

(*Continued*)

**Table 2.** (Continued)

| Variable | N (%) |
|---|---|
| Depression Score (CES-D)* | 209 (56.2) |

*A score of >10 indicates depression.

associated with greater odds of depression (AOR = 2.16, 95% CI: 0.21–0.83) and fair/poor health (AOR = 1.83, 95% CI:1.05–3.20), whereas a higher score in financial planning (*plan*) was associated with lower odds of both depression and fair/poor health (AOR = 0.42, 95% CI:0.21–0.83 and AOR = 0.46, 95% CI:0.24–0.87). There were no significant associations found with either the *spend* or *save* domains. None of the covariates were themselves associated with either having significant depressive symptoms or fair/poor physical health (results not shown).

## Internal validity

The financial health components demonstrated both discriminant and convergent validity when compared to individual measures of economic hardship (See Table 5). The *save* component was negatively correlated with household food insecurity (r = -0.124, p = 0.04) and positively correlated with wages (r = 0.423, p = 0.003). Conversely, the *borrow* component was positively correlated with food (r = 0.245, p<0.0001) and energy insecurity (r = 0.167, p = 0.007). The *spend* component was similarly correlated with energy insecurity (r = 0.311, p<0.0001). Financial health overall was negatively correlated with energy insecurity (r = -0.166, p = 0.007).

## Discussion

Findings demonstrated that financial health is distinct from other measures of economic security, such that behavior and planning indicate more of a family's ability to be financially

**Table 3. Principal components analysis (N = 373) from baseline surveys (Oct 2015 –June 2018).**

| Variable | Survey Question Text | Four Components of Financial Health | | | |
|---|---|---|---|---|---|
| | | Spend | Save | Borrow | Plan |
| Utility Shut-off | *In the last 3 months, has PGW, PECO, or other company [shut off/oil company refused to deliver] the [gas/ electricity/ oil] for not paying bills?* | **0.651*** | 0.083 | -0.191 | -0.108 |
| No Heat | *In the last 3 months, were there any days that the home was not heated or cooled because you couldn't pay the bills?* | **0.598*** | -0.079 | 0.154 | 0.154 |
| Checking | *Do you have a checking account?* | 0.042 | **0.411*** | -0.003 | 0.053 |
| Savings | *Do you have a savings account?* | -0.02 | **0.424*** | 0.007 | -0.087 |
| Direct deposit | *Do you currently have an automatic deposit or electronic transfer set up to put money away for a future use (such as savings)?* | -0.01 | **0.378*** | -0.021 | 0.037 |
| Owe money | *Do you owe anyone money (please include banks, friends, family, payday lenders, etc)?* | 0.004 | -0.036 | **0.568*** | -0.084 |
| Late Fee | *Over the last 2 months, have you paid a late fee on a loan or bill?* | -0.066 | 0.022 | **0.569*** | 0.07 |
| Financial Goal | *Do you currently have at least one financial goal?* | 0.029 | -0.039 | 0.118 | **0.667*** |
| Budget | *Do you currently have a personal budget, spending plan, or financial plan?* | 0.007 | 0.058 | -0.174 | **0.551*** |
| Spend >Save | *Over the past month, would you say your family's spending on living expenses was less than its total income?* | -0.026 | -0.02 | 0.021 | -0.026 |

* Indicates variable with an absolute value factor loading greater than 0.35. Variables with factor loading >0.35 were included in final construction of financial health domains.

**Table 4. Associations between physical and mental health outcomes and financial health domains (N = 373)[*].**

| | Presence of Significant Depressive Symptoms[a] | Fair/Poor Physical Health[b] |
|---|---|---|
| | AOR (95% CI) | AOR (95% CI) |
| Spend | 1.08 (0.68–1.70) | 1.04 (0.69–1.57) |
| Save | 0.73 (0.39–1.36) | 0.94 (0.52–1.71) |
| Borrow | **2.16 (1.21–3.87)** | **1.83 (1.05–3.20)** |
| Plan | **0.42 (0.21–0.83)** | **0.46 (0.24–0.87)** |

[*] Results are presented as adjusted odds ratios (AOR) with 95% confidence intervals in parentheses. Bolded results are statistically significant (p<0.05).

[a.] AOR represents the odds of depression associated with a unit increase in scores for each financial health domain (spend, save, borrow, or plan), adjusting for included age, race/ethnicity, educational attainment, employment, marital status, food insecurity, energy insecurity and housing insecurity.

[b.] AOR represents the odds of fair/poor physical health (vs. good/excellent health) associated with a unit increase in scores for each financial health domain (spend, save, borrow, or plan), adjusting for included age, race/ethnicity, educational attainment, employment, marital status, food insecurity, energy insecurity and housing insecurity.

'healthy' and have better physical and mental health outcomes than other individual measures of income poverty such as housing/food insecurity or income level. Results also show that domains of financial health can be defined and measured, and that they are associated with both physical and mental health. Components measured in The Network aligned closely and intuitively with a validated and established measure of financial health from the literature; these findings corroborate that there are four domains to financial health: *spend*, *save*, *borrow* and *plan*. Moreover, the *borrow* and *plan* components were associated with self-rated health and depressive symptoms, independent of food, housing, and energy security, providing further evidence that financial health may be a standalone social determinant of health.

With the PCA, questions aligned as expected, and closely mirrored indicators from CSFI. However, the question of "would you say your family's spending on living expenses was less than its total income" closely aligned with a CFSI spending indicator but not with other questions within the PCA. Associations between the financial health measure and the domains of health were mostly as anticipated. Overall, two of the domains (*borrow*, and *plan*) were associated with both meeting the cut off for depression (CES-D score >10) and self-rated physical health. This association with poor physical health is consistent with other research [56, 57]. Additionally, results from the Spearman correlations provide evidence of construct validity, distinguishing this overall financial health construct as separate from other forms of economic hardship, including housing and food insecurity.

**Table 5. Spearman correlations between financial health and other measures of economic hardship.**

| | SPEND | SAVE | BORROW | PLAN | Financial Health Overall |
|---|---|---|---|---|---|
| | R (p-value) | R (p-value) | R (p-value) | R (p-value) | R (p-value) |
| Hourly Wage | .002 (0.99) | **0.42 (.003)** | -0.01 (0.96) | .21 (0.15) | 0.04 (0.80) |
| Food Insecurity | 0.01 (0.85) | **-0.12 (0.04)** | **0.42 (< .0001)** | -0.09 (0.14) | -0.06 (0.32) |
| Energy Insecurity | **0.31 (< .0001)** | -0.02 (0.81) | **0.17 (0.007)** | 0.08 (0.19) | **-0.17 (0.007)** |
| Housing Insecurity | -0.04 (0.55) | -0.10 (0.11) | -0.04 (0.52) | -0.11 (0.06) | 0.004 (0.94) |

[*]Bolded values are significant at p<0.05.

This study has some limitations. First, causal interpretations cannot be drawn as this study is observational. Second, this study relies on self-reported information for depression and health rather than a clinical diagnosis. However, both scales have shown reliability in multiple settings and study populations. Respondents also may have underreported on socially undesirable financial questions such as paying a late fee, though this is unlikely as members were enrolling into a financial self-empowerment program. Additionally, the financial questions, structured as yes/no, may not be sufficiently precise to characterize financial health. For example, simply answering yes to "have you paid a late fee on a bill or a loan?" does not distinguish between missing one bill, paying just one late fee, and being behind on multiple bills. Also, results of the PCA are influenced by variables that possess higher degrees of both variance and associated covariance. That is, variables with large variances have important structure, while those with lower variances represent noise, a strong and sometimes incorrect assumption especially in data derived from a homogenous sample. Lastly, while using a non-parametric method like PCA is advantageous in that the output is unique and independent of the user, the fact that PCA is "agnostic to the source of the data" is also a limitation as it does not take a-priori knowledge into account as do parametric methods [58].

Despite these limitations, this study is the first to connect components of financial health drawn from an industry-backed tool to salient public health outcomes. For example, Netemeyer, et al's [38] work on financial well-being was published within a consumer research journal, yet they did not address how these perceptions are associated with specific health outcomes. Similarly, ProsperityNow, an organization devoted to advancing asset building, recently established a Financial Well-being Scale [59], yet the population which informed their scale were middle-to-high income.

While not generalizable to the general banking population, this study adds to the evidence base by using data derived from a historically underrepresented population: African-American caretakers participating in public assistance who would otherwise be missed in most conventional finance studies. Additionally, existing conceptualizations of financial health tend to focus on short-term risks and exposure to financial shocks, emphasizing savings and assets. However, most households with few or no assets are vulnerable to economic shocks [60], so considering assets alone is incomplete, and people with low incomes may rely on the alternative financial industry rather than bank accounts and partake in an informal economy where conventional forms of assets cannot be measured [45, 60, 61]. Moreover, financial studies typically rely on benchmarks unattainable for many low-income individuals, especially those that supplement income with public assistance supports such as TANF, SNAP, or the Special Supplemental Nutrition Program for Women, Infants, and Children (WIC). This study, as one of the first to assess financial health among low-income caregivers, provides a foundation for future research that can enable public health practitioners and researchers to further understand the relationship between financial health and physical health among caregivers of young children. Additionally, as social service providers can measure patient and client financial health, they can more appropriately provide resources and solutions to help low-income families improve their overall health. Providing early interventions to help caregivers of young children to improve financial health can benefit children with lifelong health and wellbeing.

## Practice implications

The American Psychological Association considers financial stress to be one of the top stressors in America [62], with severe stresses considered an adversity akin to adverse childhood experiences (ACEs), which include abuse, neglect, and household dysfunction [63]. Financial stress and economic hardships have been linked with increased physical pain, lowered pain

tolerance, and risk of coronary heart disease [64]. Despite this evidence, comprehensive efforts to consider financial health as a mode of fostering financial capability and asset building that: 1) protect from the very financial shocks that can lead to or exacerbate economic hardship; and 2) address the harmful effects of the disparities in income and wealth [65–67], are missing from the public health landscape [14–17].

In order to better understand financial health and its implications for public health, consistent definition and measurement are necessary. Integrating financial health into the overall understanding of health determinants will help guide more effective and far-reaching interventions by providing a framework for solutions that do not rely on isolated approaches to food, housing, energy or other economic insecurities, but rather can address all of these issues by seeking to improve financial health in its four domains of *spend*, *save*, *borrow* and *plan*. Moreover, to reduce health disparities and address root causes of poor health, the public health community should generate opportunities for families who are financially and socially marginalized to build their wealth. This work sets the foundation for future researchers to investigate and agree upon a consistent measure and definition of financial health, which may help to generate interventions that build people's capacities to improve their financial situation and may help inform policies that can increase or eliminate asset limits, promoting savings, and ultimately promote wellbeing.

## Supporting information

**S1 Data.**
(SAS7BDAT)

## Author Contributions

**Conceptualization:** Emily Brown Weida, Pam Phojanakong, Mariana Chilton.

**Formal analysis:** Pam Phojanakong.

**Funding acquisition:** Mariana Chilton.

**Investigation:** Emily Brown Weida, Mariana Chilton.

**Methodology:** Emily Brown Weida, Pam Phojanakong.

**Project administration:** Emily Brown Weida, Falguni Patel, Mariana Chilton.

**Resources:** Mariana Chilton.

**Supervision:** Mariana Chilton.

**Writing – original draft:** Emily Brown Weida, Pam Phojanakong.

**Writing – review & editing:** Emily Brown Weida, Pam Phojanakong, Falguni Patel, Mariana Chilton.

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
