## [Decision Letter · Decision Letter 0]

14 Jan 2020

PONE-D-19-31725

Financial Health as a Measurable Social Determinant of Health

PLOS ONE

Dear Mrs. Weida,

Thank you for submitting your manuscript to PLOS ONE. After careful consideration, we feel that it has merit but does not fully meet PLOS ONE’s publication criteria as it currently stands. Therefore, we invite you to submit a revised version of the manuscript that addresses the points raised during the review process.

The paper has a great deal of promise, but it needs some revisions. First, there is a lack of supporting literature in the introduction. Please add more information about the history of finance as a component of overall health. If it is postulated to be an interdisciplinary perspective on health, this needs to be more clearly stated. The analyses should be more sophisticated given the robustness of the data. Please note the suggestions made by the reviewers (below). 

We would appreciate receiving your revised manuscript by Feb 28 2020 11:59PM. To enhance the reproducibility of your results, we recommend that if applicable you deposit your laboratory protocols in protocols.io, where a protocol can be assigned its own identifier (DOI) such that it can be cited independently in the future. For instructions see: http://journals.plos.org/plosone/s/submission-guidelines#loc-laboratory-protocols

We look forward to receiving your revised manuscript.

Kind regards,

Heidi H Ewen

Academic Editor

PLOS ONE

Journal Requirements:

2. Please ensure you have cited any additional publications that have arisen from the Building Wealth and Health Network Trial in the manuscript.

Reviewers' comments:

Reviewer's Responses to Questions

**Comments to the Author**

1. Is the manuscript technically sound, and do the data support the conclusions?

Reviewer #1: Partly

Reviewer #2: Yes

2. Has the statistical analysis been performed appropriately and rigorously? 

Reviewer #1: No

Reviewer #2: No

3. Have the authors made all data underlying the findings in their manuscript fully available?

Reviewer #1: No

Reviewer #2: Yes

4. Is the manuscript presented in an intelligible fashion and written in standard English?

Reviewer #1: Yes

Reviewer #2: Yes

5. Review Comments to the Author

Reviewer #1: Reviewer’s Comments for “Financial Health as a Measurable Social Determinant of Health”

It was a pleasure to read this manuscript in that the topic of this study is timely important and worth to be explored beyond the discipline. Nevertheless, I would bring up some questions and concerns that may help the manuscript strengthened.

Title

1. This study is not to examine the relationship between “financial health” and “health” because financial health variables are not used as predictors or determinants of health. Thus, the title may mislead readers. The authors should conduct further analyses to use this title.

Abstract

2. “Objectives” does not get the gist. Is the purpose of this study to develop the new measurement of financial health? Or to suggest a root solution for improving financial health?

Keywords

3. Again, this study does not reflect the “social determinant of health.” Be careful to include it.

Introduction

4. I can see the concept, “financial capability” throughout the manuscript. What is the difference between financial health and financial capability? They sound similar.

5. P3 Line 69: class � does it mean social class?

Methods

Parent Study and Study Sample

6. Please provide a bit more explanation about the Building Wealth and Health Network program. Readers have less information on what this program is (e.g., what organization develops the program, where or in what area/region it is run, how many people have participated in this program).

7. It would be better to use the term “participants” instead of “members” of the program.

8. As authors are stating about the “study sample,” elucidate the total sample used in this study.

9. P5 Line 103: <6yrs � please spell out and write in a full sentence.

Economic Hardship and Financial Measures

10. There is an unmatched question:

P6 Line135: In the last three months I repaid the money I owned on time

Table 1. (Borrow x The Network Survey Questions): Within the last six months, I repaid the money I owned on time

11. Please say more about how you code variables that align with the four categories. What answers are considered as affirmative or negative? Which variables were reversely coded? Does a higher/lower score in each category (i.e., spend, save, borrow, plan) represent a higher level of financial health?

12. Where does the Financial Behavior, Knowledge, and Self-Efficacy Scale come from? This a new term/concept that you need to explain.

Statistical Analysis

13. P8 Line 158: Are economic hardship, food insecurity, energy insecurity, and housing insecurity all different concepts?

14. P8 Line 169: Why do authors use .35 as the cutoff value for factor loadings? If there is a reference, please cite it.

15. The sample size of this study is 373, which may fulfill the normality assumption. Please add more explanation of why authors used a nonparametric test (i.e., Mann-Whitney U Test) instead of t-test.

16. P8 Line 161: Specify which energy security question is included (or excluded) in the analysis.

Result

17. Table 2: How household food security, Hosing insecurity, and Energy insecurity are measured? If they are indicating the questions used in Table 1, you need to specify which one is which.

18. Table 3: Please complete the table. What do the first and second columns stand for?

19. It would be better to use the term “participants” instead of “members” of the program.

20. Table 4: Mann-Whitney U Test is used to compare the mean value differences between two independent groups. Thus, you cannot use the term “Association.”

Internal Validity

21. Please add tables for this part of the result. Also, authors need to provide more explanation about food insecurity, energy insecurity, and wages. Are food insecurity and energy insecurity subordinate to economic hardship? How are they measured?

22. P13 Line 224-225: what are “other distinct constructs of economic hardship”? How can you say financial health is distinct from other distinct constructs of economic hardship? Is it tested?

23. Where are the results for the Spearman rank test?

Discussion

24. Again, Mann-Whitney U Test is not testing the association between two variables. Thus, you cannot say “the associations between our financial health measure and the domains of health were as we anticipated.” If you’d like to see the association or relationship between those variables, additional statistical tests are required (e.g., regression).

25. The section explaining Figure 1 is missing.

26. Overall, check verb tenses. Some are used in present tenses, and some are used in past tenses.

Reviewer #2: PLOUS-D-19-31725

Paper Review

This paper was well written and nicely conceptualized. I must say, however, that I was surprised a bit by the framing of the paper. The notion of financial health is really not a new concept. Financial health has its origins in the 1970s and 1980s. What has happened, unfortunately, is that health researchers, in general, have not gone back deeply enough when conducting literature reviews to find this early work. Much of the early work was published as family resource management, consumer economics, family economics, or under the broader categorization of family and consumer sciences. Beginning in 1990, the work on financial health transitioned to the field of financial counseling and planning. This paper references a few papers from the Journal of Financial Counseling and Planning; it is this type of journal where much of the early work was being published. The leading researchers in the field include O’Neill, Kim, and Xiao; however, their work is seldom published in health journals, which explains why health researchers often assume that there is a paucity of research in the area. Here is an example of the type of work that exists, but which is rarely cited in health journals:

• Small steps to health and wealth™: Program update and research insights B O'Neill, K Ensle - The Forum for Family and Consumer Issues, 2014 - theforumjournal.org

• Propensity to plan: A key to health and wealth

B O'Neill, JJ Xiao, K Ensle - Journal of Financial Planning, 2016 - search.proquest.com

• Financially distressed consumers: Their financial practices, financial well-being, and health

B O'Neill, B Sorhaindo, JJ Xiao… - Journal of Financial …, 2005 - papers.ssrn.com

• Changes in health, negative financial events, and financial distress/financial well-being for debt management program clients

B O'Neill, A Prawitz, B Sorhaindo, J Kim… - Journal of Financial …, 2006 - papers.ssrn.com

• The small steps to health and wealth™ challenge: An online tool to motivate consumers to make positive behavior changes

B O'Neill, K Ensle - Proceedings of the Eastern Family Economics …, 2010 - fermascholar.org

• Family health and financial literacy–forging the connection

B Braun, J Kim, EA Anderson - Journal of Family and Consumer …, 2009 - sph.umd.edu

Anyway, this is just a summary of my initial reactions.

Comments:

I would like more information on the measure used in the study. The questions seem very similar to scaling items developed by Garman and his students Joo and Kim (e.g., InCharge Financial Distress/Financial Well-Being Scale: Development, Administration, and Score Interpretation).

As far as I know, this was one of the first dissertations on the topic of financial stressors, which is essentially what is being measured in the “new” scale: https://vtechworks.lib.vt.edu/handle/10919/30519 It is possible that the CFSI measure was developed totally independently of other work related to financial stress and financial stressors, but that seems somewhat unlikely given that any google scholar search would bring up some important past research. It is possible that had a literature review been added to the current paper, much of my concern/questioning would be addressed.

A few questions/issues:

1. Is there a reason the paper is missing a review of literature? (The answer may be that it is not common for this journal to publish reviews of literature as an element of an accepted paper.)

2. On p. 12, was the spend>save question included in any of the domains? The factor loadings are very small across the domains. If it was not included, why is it shown? If it was included, why?

3. Given the sample size, and the demographic data available, why was the choice made to use a non-parametric bivariate test rather than a multivariate model? The literature shows that several demographic variables are typically associated with health and wealth outcomes. It seems like the Mann-Whitney U test would be the first stage level of analysis followed by a MANOVA or even a series of regressions.

I do like how the authors have concluded that integrating financial health into a broader definition of health is important. It may take more papers like this to help shift the health research to this acknowledgement. I like the paper; my general conclusion is that this seems very brief given the topic; also, I do think a more robust series of tests would help the paper quite a bit.

6. PLOS authors have the option to publish the peer review history of their article (what does this mean?). If published, this will include your full peer review and any attached files.

Reviewer #1: No

Reviewer #2: No

---

## [Author Response · Author response to Decision Letter 0]

24 Mar 2020

Thank you for the great suggestions and edits to strengthen our manuscript. For a clearer point by point response, please see our attached document "Response to Reviewers". 

General Comments: 

The paper has a great deal of promise, but it needs some revisions. First, there is a lack of supporting literature in the introduction. Please add more information about the history of finance as a component of overall health. If it is postulated to be an interdisciplinary perspective on health, this needs to be more clearly stated. The analyses should be more sophisticated given the robustness of the data. Please note the suggestions made by the reviewers (below). 

Thank you for this feedback. We have improved the introduction and added more background literature Please see lines 65-118.

 Thank you for catching this, we have now adhered to PLOS ONE’s file naming requirements. 

Please ensure you have cited any additional publications that have arisen from the Building Wealth and Health Network Trial in the manuscript. 

Thank you for this reminder. We have added this into the text, please see line 135. 

In your Data Availability statement, you have not specified where the minimal data set underlying the results described in your manuscript can be found. PLOS defines a study's minimal data set as the underlying data used to reach the conclusions drawn in the manuscript and any additional data required to replicate the reported study findings in their entirety. All PLOS journals require that the minimal data set be made fully available

We have included a minimal de identified dataset as a Supporting Information file. 

Please include a separate caption for each figure in your manuscript. 

We have ensured that our figure has a separate caption within the text.

Reviewer 1: It was a pleasure to read this manuscript in that the topic of this study is timely important and worth to be explored beyond the discipline. Nevertheless, I would bring up some questions and concerns that may help the manuscript strengthened.

Title

1. This study is not to examine the relationship between “financial health” and “health” because financial health variables are not used as predictors or determinants of health. Thus, the title may mislead readers. The authors should conduct further analyses to use this title. 

Thank you for this comment. We have conducted logistic regressions with financial health domains as predictors of our health outcomes to better reflect the title and objectives of the paper.

Abstract

2. “Objectives” does not get the gist. Is the purpose of this study to develop the new measurement of financial health? Or to suggest a root solution for improving financial health? 

Thank you for this point. We updated our objectives to better reflect the purpose of the paper.

Keywords

3. Again, this study does not reflect the “social determinant of health.” Be careful to include it. 

Thank you for this catch. We have updated our analyses to reflect the use of this term more appropriately. 

Introduction

4. I can see the concept, “financial capability” throughout the manuscript. What is the difference between financial health and financial capability? They sound similar. 

Thank you for the opportunity to clarify. Financial capability relates more to an individual’s knowledge and skills to make money-related decisions whereas we are claiming that financial health is a more holistic approach to a family’s financial situation. We have included this definition for financial capability to the introduction to help clarify.

5. P3 Line 69: class � does it mean social class? 

The term class is out of place and has been removed. 

Methods

Parent Study and Study Sample

6. Please provide a bit more explanation about the Building Wealth and Health Network program. Readers have less information on what this program is (e.g., what organization develops the program, where or in what area/region it is run, how many people have participated in this program). 

Thank you for pointing out this oversight. We have added in text to reflect the number of participants from the pilot (N=103) as well as the Phase II (N=373), and that the study takes place in Philadelphia, PA out of Drexel University. 

7. It would be better to use the term “participants” instead of “members” of the program 

Thank you for this opportunity to clarify why we use the term “members” over “participants”. All participants in The Network are referred to as “members” to reflect the peer format nature of the program to foster social capital, connection and belonging. We have added these citations to the text (please see line 126)

8. As authors are stating about the “study sample,” elucidate the total sample used in this study. 

We have specified that the sample was taken from all participants recruited for Phase II at baseline (lines 196-197).

9. P5 Line 103: <6yrs � please spell out and write in a full sentence. 

Thank you. We have updated this in the text.

Economic Hardship and Financial Measures

10. There is an unmatched question:

P6 Line135: In the last three months I repaid the money I owned on time

Table 1. (Borrow x The Network Survey Questions): Within the last six months, I repaid the money I owned on time 

Thank you for catching this. The question should read “Within the last three months I repaid the money I owed on time.” We revised and updated the table. 

11. Please say more about how you code variables that align with the four categories. What answers are considered as affirmative or negative? Which variables were reversely coded? Does a higher/lower score in each category (i.e., spend, save, borrow, plan) represent a higher level of financial health? 

Thank you for this opportunity to clarify. Please see Lines 169-172 and Table 1.

12. Where does the Financial Behavior, Knowledge, and Self-Efficacy Scale come from? This a new term/concept that you need to explain. 

Thank you for this comment. This scale comes from Danes et al. (1999); as this study is a financial empowerment study we used this scale to help assess our members’ post programmatic growth in financial behaviors, knowledge and self-efficacy. We have included a brief description in the text please see lines 152-153. 

Statistical Analysis

13. P8 Line 158: Are economic hardship, food insecurity, energy insecurity, and housing insecurity all different concepts? 

Thank you for catching this – we have placed parentheticals around food insecurity, energy insecurity and housing insecurity to more clearly indicate these are forms of economic hardship. 

14. P8 Line 169: Why do authors use .35 as the cutoff value for factor loadings? If there is a reference, please cite it. 

Line 210, Cite: Peres-Neto, P. R., Jackson, D. A., & Somers, K. M. (2003). Giving meaningful interpretation to ordination axes: assessing loading significance in principal component analysis. Ecology, 84(9), 2347-2363.

15. The sample size of this study is 373, which may fulfill the normality assumption. Please add more explanation of why authors used a nonparametric test (i.e., Mann-Whitney U Test) instead of t-test. 

Thank you for this opportunity to clarify. As we have decided to enhance the robustness of our analyses, we have removed the Mann-Whitney U Test and instead have included a series of regressions. 

16. P8 Line 161: Specify which energy security question is included (or excluded) in the analysis. 

We apologize for the confusion. While the questions are from a validated energy insecurity screener, we chose to use them in our analysis as indicators of being unable to pay bills on time and have clarified this in the manuscript (lines 156-157).

Result

17. Table 2: How household food security, Hosing insecurity, and Energy insecurity are measured? If they are indicating the questions used in Table 1, you need to specify which one is which. 

Thank you for this point. We have added in descriptions on these measurements. Please see lines 173-181.

18. Table 3: Please complete the table. What do the first and second columns stand for? 

Thank you for catching this oversight. We have added column headers for this table. 

19. It would be better to use the term “participants” instead of “members” of the program. 

Please see our response above. All participants in The Network are referred to as “members” to reflect the peer format nature of the program to foster social capital, connection and belonging. We have added this text (please see lines 126)

20. Table 4: Mann-Whitney U Test is used to compare the mean value differences between two independent groups. Thus, you cannot use the term “Association.” 

Thank you for pointing out the error in our language; we have since replaced the Mann-Whitney U tests with regression analyses. 

Internal Validity

21. Please add tables for this part of the result. Also, authors need to provide more explanation about food insecurity, energy insecurity, and wages. Are food insecurity and energy insecurity subordinate to economic hardship? How are they measured? 

Thank you for this suggestion. We have included the table, and we have provided citations for the food and energy insecurity measures, as well as provided more explanation for Figure 1, which depicts food and energy insecurity as types of economic hardship. 

22. P13 Line 224-225: what are “other distinct constructs of economic hardship”? How can you say financial health is distinct from other distinct constructs of economic hardship? Is it tested? 

Thank you for this point. As it has not been tested, we have removed this statement. 

23. Where are the results for the Spearman rank test? 

We have removed the Spearman and Mann-Whitney tests to make room for the logistic regressions to enhance the robustness of our analyses. 

24. Again, Mann-Whitney U Test is not testing the association between two variables. Thus, you cannot say “the associations between our financial health measure and the domains of health were as we anticipated.” If you’d like to see the association or relationship between those variables, additional statistical tests are required (e.g., regression). 

We have added logistic regression models to our analyses that examine the odds of depression and poor health with each of our financial health domains while adjusting for demographics and other “established” determinants of health to show that there is an independent association between health and aspects of financial health.

25. The section explaining Figure 1 is missing. 

Thank you for pointing out this oversight. We have added text (see lines 96-98) explaining Figure 1. 

26. Overall, check verb tenses. Some are used in present tenses, and some are used in past tenses. 

Thank you for this comment. We have gone through and hopefully corrected these inconsistencies. 

Reviewer #2: 

This paper was well written and nicely conceptualized. I must say, however, that I was surprised a bit by the framing of the paper. The notion of financial health is really not a new concept. Financial health has its origins in the 1970s and 1980s. What has happened, unfortunately, is that health researchers, in general, have not gone back deeply enough when conducting literature reviews to find this early work. Much of the early work was published as family resource management, consumer economics, family economics, or under the broader categorization of family and consumer sciences. Beginning in 1990, the work on financial health transitioned to the field of financial counseling and planning. This paper references a few papers from the Journal of Financial Counseling and Planning; it is this type of journal where much of the early work was being published. The leading researchers in the field include O’Neill, Kim, and Xiao; however, their work is seldom published in health journals, which explains why health researchers often assume that there is a paucity of research in the area. Here is an example of the type of work that exists, but which is rarely cited in health journals: 

Thank you for these comments and for providing examples of literature to include. We have updated our literature review. 

I would like more information on the measure used in the study. The questions seem very similar to scaling items developed by Garman and his students Joo and Kim (e.g., InCharge Financial Distress/Financial Well-Being Scale: Development, Administration, and Score Interpretation).

As far as I know, this was one of the first dissertations on the topic of financial stressors, which is essentially what is being measured in the “new” scale: https://vtechworks.lib.vt.edu/handle/10919/30519 It is possible that the CFSI measure was developed totally independently of other work related to financial stress and financial stressors, but that seems somewhat unlikely given that any google scholar search would bring up some important past research. It is possible that had a literature review been added to the current paper, much of my concern/questioning would be addressed. 

Thank you for this question, we have provided a review of the literature to hopefully provide some clarity. We have included some of the great work done by Garman and his colleagues in our literature review. 

1. Is there a reason the paper is missing a review of literature? (The answer may be that it is not common for this journal to publish reviews of literature as an element of an accepted paper.) 

Thank you for catching this. We have provided a review of the literature. Please see lines 77-118.

2. On p. 12, was the spend>save question included in any of the domains? The factor loadings are very small across the domains. If it was not included, why is it shown? If it was included, why? 

The spend save question was not included in the final construction of any of the domains but was included in the initial PCA as a potentially relevant variable. We chose to include it in the table to be transparent about what variables were considered for the PCA, even if they were not statistically useful at the end. To avoid confusion, we have added text to the footnote to specify which variables contributed to each domain.

3. Given the sample size, and the demographic data available, why was the choice made to use a non-parametric bivariate test rather than a multivariate model? The literature shows that several demographic variables are typically associated with health and wealth outcomes. It seems like the Mann-Whitney U test would be the first stage level of analysis followed by a MANOVA or even a series of regressions. 

Thank you for this suggestion. We have added multivariate logistic regression models to our analyses.

I do like how the authors have concluded that integrating financial health into a broader definition of health is important. It may take more papers like this to help shift the health research to this acknowledgement. I like the paper; my general conclusion is that this seems very brief given the topic; also, I do think a more robust series of tests would help the paper quite a bit. 

Thank you for this comment, as well as your suggestions. We hope that our added literature review and additional analyses have strengthened our paper.

---

## [Decision Letter · Decision Letter 1]

10 Apr 2020

PONE-D-19-31725R1

Financial Health as a Measurable Social Determinant of Health

PLOS ONE

Dear Ms. Weida,

Thank you for submitting your manuscript to PLOS ONE. After careful consideration, we feel that it has merit but does not fully meet PLOS ONE’s publication criteria as it currently stands. Therefore, we invite you to submit a revised version of the manuscript that addresses the points raised during the review process.

Both of the reviewers have noted some copy-editing issues that need to be addressed before the paper is ready for publication. Once these edits have been made, I will review the resubmission for final determination.

We would appreciate receiving your revised manuscript by May 25 2020 11:59PM. To enhance the reproducibility of your results, we recommend that if applicable you deposit your laboratory protocols in protocols.io, where a protocol can be assigned its own identifier (DOI) such that it can be cited independently in the future. For instructions see: http://journals.plos.org/plosone/s/submission-guidelines#loc-laboratory-protocols

We look forward to receiving your revised manuscript.

Kind regards,

Heidi H Ewen, Ph.D.

Academic Editor

PLOS ONE

Reviewers' comments:

Reviewer's Responses to Questions

**Comments to the Author**

1. If the authors have adequately addressed your comments raised in a previous round of review and you feel that this manuscript is now acceptable for publication, you may indicate that here to bypass the “Comments to the Author” section, enter your conflict of interest statement in the “Confidential to Editor” section, and submit your "Accept" recommendation.

Reviewer #1: All comments have been addressed

Reviewer #2: All comments have been addressed

2. Is the manuscript technically sound, and do the data support the conclusions?

Reviewer #1: Yes

Reviewer #2: Yes

3. Has the statistical analysis been performed appropriately and rigorously? 

Reviewer #1: Yes

Reviewer #2: Yes

4. Have the authors made all data underlying the findings in their manuscript fully available?

Reviewer #1: Yes

Reviewer #2: Yes

5. Is the manuscript presented in an intelligible fashion and written in standard English?

Reviewer #1: Yes

Reviewer #2: Yes

6. Review Comments to the Author

Reviewer #1: The manuscript has been considerably improved and sophisticated. I have a few questions as follows:

1. Research question and Discussion

The answer to the first research question (i.e., introduce a construct of financial health into the domain of public health defined by more than just individual measures of economic hardship) is not clearly stated in the discussion section. Please add more explanations why and which part of your analyses can be the evidence of the distinction between financial health and economic hardship (indicated by food security, housing security, and energy security).

2. “To avoid collinearity due to the interrelated nature of energy security, housing security and food security, only questions relating to energy security were used for the PCA (p.10, line 119-201)”. If only energy security items were included in the PCA, how do we know financial health variables are distinct from housing security and food security?

3. As the sample represents low-income caregivers of young children, it would be more valuable if the discussion provides implications beneficial to this specific population.

Reviewer #2: Thank you for addressing my issues. The paper does need a bit of copy editing. This may be something the editorial office can help with (e.g., some of the discussion around the new logistic regression model was oddly written). Other than that, the additions to the paper were appropriate.

7. PLOS authors have the option to publish the peer review history of their article (what does this mean?). If published, this will include your full peer review and any attached files.

Reviewer #1: No

Reviewer #2: No

---

## [Author Response · Author response to Decision Letter 1]

1 May 2020

Reviewer #1: The manuscript has been considerably improved and sophisticated. I have a few questions as follows:

1. Research question and Discussion

The answer to the first research question (i.e., introduce a construct of financial health into the domain of public health defined by more than just individual measures of economic hardship) is not clearly stated in the discussion section. Please add more explanations why and which part of your analyses can be the evidence of the distinction between financial health and economic hardship (indicated by food security, housing security, and energy security).

Thank you for the opportunity to provide more clarification. We added text in the discussion (please see lines 295-297). 

2. “To avoid collinearity due to the interrelated nature of energy security, housing security and food security, only questions relating to energy security were used for the PCA (p.10, line 119-201)”. If only energy security items were included in the PCA, how do we know financial health variables are distinct from housing security and food security? 

Thank you for this insight. The Spearman correlation tests between each domain of our financial health construct (as well as the overall score) and housing and food insecurity were provided as demonstration construct validity to show that financial health is distinct from housing and food insecurity. The text that we added in response to your previous question will likely help us to clarify for readers the intention of these results. 

3. As the sample represents low-income caregivers of young children, it would be more valuable if the discussion provides implications beneficial to this specific population.

 Thank you for this point. Please see lines 321-330. We have added narrative at the end of the discussion section (lines 332-339) that describes the implications of this study to benefit low-income families with young children.

Reviewer #2: Thank you for addressing my issues. The paper does need a bit of copy editing. This may be something the editorial office can help with (e.g., some of the discussion around the new logistic regression model was oddly written). Other than that, the additions to the paper were appropriate. 

Thank you for the opportunity to correct these, we have taken time for more careful copyediting.

---

## [Editor Report · Decision Letter 2]

5 May 2020

Financial Health as a Measurable Social Determinant of Health

PONE-D-19-31725R2

Dear Dr. Weida,

We are pleased to inform you that your manuscript has been judged scientifically suitable for publication and will be formally accepted for publication once it complies with all outstanding technical requirements.

With kind regards,

Heidi H Ewen, Ph.D.

Academic Editor

PLOS ONE
---

## [Editor Report · Acceptance letter]

8 May 2020

PONE-D-19-31725R2 

Financial Health as a Measurable Social Determinant of Health 

Dear Dr. Weida:

I am pleased to inform you that your manuscript has been deemed suitable for publication in PLOS ONE. Congratulations! Your manuscript is now with our production department. 

With kind regards,

on behalf of

Dr. Heidi H Ewen 

Academic Editor

PLOS ONE